# Studies on Flavor Compounds and Free Amino Acid Dynamic Characteristics of Fermented Pork Loin Ham with a Complex Starter

**DOI:** 10.3390/foods11101501

**Published:** 2022-05-21

**Authors:** Zhiqing Tian, Qiujin Zhu, Yuanshan Chen, Ying Zhou, Ke Hu, Hongying Li, Kuan Lu, Jie Zhou, Yuan Liu, Xi Chen

**Affiliations:** 1School of Liquor and Food Engineering, Guizhou University, Guiyang 550005, China; zqtian1996@163.com (Z.T.); yschen20220423@163.com (Y.C.); zhying_0525@163.com (Y.Z.); khu3@gzu.edu.cn (K.H.); zj707211765@163.com (J.Z.); 2Laboratory of Animal Genetics, Breeding and Reproduction in the Plateau Mountainous Region, Ministry of Education, Collaborative Innovation Center for Mountain Ecology & Agro-Bioengineering (CICMEAB), College of Life Sciences, Guizhou University, Guiyang 550005, China; hongyingli_718@163.com (H.L.); wukong4608@163.com (K.L.); 3Department of Food Science & Technology, School of Agriculture & Biology, Shanghai Jiao Tong University, Shanghai 200240, China; y_liu@sjtu.edu.cn; 4China Meat Research Center, Beijing 100068, China; 961983@163.com

**Keywords:** GC-MS, fermented loins, FAA, dynamic characteristics

## Abstract

*Staphylococcus simulans* and *Lactobacillus plantarum* screened from Guizhou specialty food were used to prepare fermented pork loin ham. The sensory qualities and flavor profiles of fermented pork loin hams from 0 to 42 days were investigated in order to reveal the dynamics of fermented pork loin ham. The results show that total free amino acids (TFAA) content reached the highest value on the 35th day, and the umami amino acids, including aspartic acid (ASP), glutamic acid (GLU), glycine (GLY), and alanine (ALA), were the main amino acids in all periods. Notably, the RV coefficient (0.875) indicates that free amino acids (FAA) are highly correlated with the sensory score of the E-tongue. In terms of the volatile compounds identified, the esters content gradually increased between 7 and 42 days, and ethyl octanoate was the most abundant compound during all periods. These esters imparted a characteristic aroma component to the fermented pork loin ham. The most important finding was that the increase in the content of esters represented by octanoic acid-ethyl ester might be related to the increase in the content of FAA with the increase in fermentation time. Both the E-nose and E-tongue showed good discrimination ability for fermented tenderloin ham with different fermentation times, which was crucial in cases with large clusters. In addition, the multiple factor analysis (MFA) indicated that the E-nose aroma value might be the key factor in distinguishing fermented pork loin ham with different fermentation times.

## 1. Introduction

Consumers recognize dry-cured pork loin due to its unique flavor [1]. There are four main ways to produce flavor in fermented meat products: protein degradation, lipid oxidation, the Maillard reaction, and microorganisms [2]. As we know, as a geographical indication product, the flavor quality of ham is inseparable from environmental microorganisms. Some reports have summarized the relationship between microorganisms and ham flavor formation [3,4]. Generally speaking, this relationship is because they can degrade proteins and lipids to form micro-molecular substances. Therefore, in some recent studies, researchers have gained interest in fermented meat using beneficial microbes from local specialty meat products as a starter, giving fermented meat more local characteristics [5,6]. According to Laranjo et al., starters, especially *Lactobacillus* and *Staphylococci*, can improve the safety of fermented meat products, help standardize product characteristics, and shorten the aging time [7]. As the most important sensory evaluation of food, the flavor of ham tends to change dramatically due to prolonged fermentation. Seong et al. reported that the flavor of pork loin ham varied significantly in the short term, but then tended to be consistent [8]. Therefore, the dynamics of pork loin ham should be explored using some modern sensory technologies.

With the development of sensory evaluation technologies, including electronic nose (E-nose), gas chromatography–mass spectrometry (GC–MS), electronic tongue (E-tongue), and so forth, these have been applied to the evaluation of food aroma value evaluation, intelligent sensing technology, and the development of combined GC-MS technology with E-nose [9,10,11,12]. These detection techniques help us to gain a better understanding of flavor composition and development. In addition, some reports suggest that the most important precursors for ham flavor development are free amino acids (FAA) [13]. FAA are formed by the degradation of peptide chains, which are formed by protein macromolecules under the action of endogenous enzymes, exogenous enzymes and microorganisms [14]. Zang et al. indicated that the control of crucial flavor precursors during fermentation was a potential method for improving the flavor of fermented fish [15].

In our previous study, *Lactobacillus plantarum* SJ4 and *Staphylococcus simulans* ZF4 were screened from Guizhou sour pork (China) and Zunyi preserved meat (China), respectively, and used as starters. This study used the longissimus of pork as the raw material to exploit the fermented pork loin ham. Based on a series of test results of E-nose, E-tongue, GC-MS, and FAA, the changes in flavor and taste of fermented pork loin ham were explored within 0–42 days. At the same time, the relationship between flavor compounds and FAA was studied, which provided a theoretical reference for the improvement in fermented loin ham technology.

## 2. Materials and Methods

### 2.1. Activation and Preparation of Starter

*S. simulans* ZF4 (the accession number of the gene sequence on Genbank is ON193993) was preserved in the laboratory, and *L. plantarum* SJ4 was preserved by the China Center of Industrial Culture Collection (strain number: 20-523-1634-2388). Mannitol salt agar medium (MSA) media were used to activate ZF4, and the Man Rogosa Sharpe Broth (MRS) medium was used to activate SJ4. After starting the strains for three generations and washing them three times with 0.9% saline, their concentrations were adjusted to 10^9^ CFU/mL for subsequent use.

### 2.2. Technological Process of Fermented Pork Loins

The fermentation process was slightly modified according to Belloch et al. [16,17]. The diet of landrace pigs was somewhat slightly adjusted in different stages, and the main ingredients were corn and wheat bran. Landrace pigs were slaughtered after being fed about 250 kg of diet, and the tenderloin was sold by Fuzhiyuan Group Co., Ltd (Guiyang, China). Spices (food-grade) were purchased from a supermarket in Guiyang City, and the casings were purchased from the Shenguan Group. The pork loin was cut into about 500 g per pork loin cubes equally (*n* = 21). The loin’s surface was rubbed with a formula containing 3% salt (*w*/*w*), 0.01% sodium nitrite, 0.2% sodium erythorbate, 0.5% sodium tripolyphosphate, 1% sugar, and 1% glucose, and kept at 4 °C for 24 h to allow the seasoning mixture to distribute into the meat. The prepared bacterial suspension was inoculated into the meat (10^7^ CFU/g), with a ZF4 to SJ4 ratio of 1:1. Afterward, the loins were stuffed into 50-mm-diameter collagen casings and immediately hung in a constant temperature and constant humidity fermentation cabinet. The fermentation cabinet was set to 30 °C and 85% RH with stable circulating air to allow the starter to work. The samples were taken every 7 days until the end of fermentation on day 42 to compare the dynamic evolution of flavor and free amino acids (FAA)at different fermentation times. Each time, three fermented tenderloin hams of the same period were minced and mixed, and then put into sterile vacuum bags and stored at −20 °C until testing.

### 2.3. E-tongue Measurement

E-tongue measurements were performed using a TS-5000Z taste-sensing system (Insent Inc., Kanagawa, Japan) equipped with five lipid membrane sensors that indicated different taste qualities and corresponding reference electrodes. The samples (60 g) were added to 240 mL of deionized water, followed by a 40 °C—water bath for 10 min before the liquid was filtered. The two-step cleaning method was used for detection, and the sample collection and cleaning were performed alternately as follows: cleaning solution 1, 90 s; cleaning solution 2, 120 s; cleaning solution 3, 120 s; conditioning solution, 30 s; sample selection, 30 s; cleaning solution 4, 3 s; cleaning solution 5, 3 s; and CPA solution, 30 s. Each sample was measured four times, with the last three measurements selected for analysis. The average value of the stable was calculated. The response signal was selected as the output value for each period of 110–120 s, and the ambient detection temperature was 25 ± 2 °C. A taste analysis was used to convert the data into gust values for data analysis. Three parallel tests were performed on the sample.

### 2.4. GC-MS Determination

Initially, 5 g of the sample was placed in a 20-mL headspace sample bottle, and equilibration of the headspace was done, maintaining the vial at room temperature (25 ± 2 °C) during 30min, as described by Perea-Sanz with minor modifications [18]. The 50/30-μm DVB/Carboxen/PDMS fiber (Supelco, Bellefonte, PA, USA) was headed into the SPME device, which was, in turn, inserted into the vial and shaken at 50 °C for 30 min to extract and absorb the volatile components. Then, it was desorbed in 7 min at 250 °C into the GC inlet with the automatic autosampler. GC-MS included the Thermo TRACE 1300, Thermo TSQ800 EVO gas chromatography triple quadrupole mass spectrometer, and DB-5 MS (30 m × 0.25 mm × 0.25 μm). The inlet temperature was 250 °C. High-purity helium with a flow rate of 1 mL/min was used as the carrier gas. The analysis was performed in the splitless mode. The initial temperature was 40 °C for 5 min; then the temperature was increased to 150 °C at 5 °C/min and maintained for 3 min; and finally, the temperature was increased to 240 °C at 5 °C/min and maintained for 3 min. The electron bombardment energy of the EI+ ion source was 70 eV, the ion source temperature was 250 °C, the transmission line temperature was 240 °C, and the mass range was 35 amu–550 amu. The GC-MS raw data was initially searched with NIST5 and then combined with the retention index. The Xcalibur software was used to determine the background correction and resolution of the peak automatically. System software was used for chemometric analysis to determine the possible compounds of the peak in the sample (RSI matching degree greater than 800).

### 2.5. FAA Determination

Initially, 1 g sample was extracted with 50 mL of 0.01N hydrochloric acid for 30 min, shaken, and filtered. Then, 2 mL of the filtrate was accurately pipetted into a centrifuge tube, and 2 mL of 8% sulfosalicylic acid was added and allowed to stand for 15 min. The supernatant was tested after passing through a 0.22-µm membrane. Sykam S-433D, including LCAK07/LI-type standard analytical cation exchange resin, was used to test 17 FAA in fermented pork loin ham. The injection volume was 50 μL. The detection wavelength used in channel 1 was 440 nm, which was used to detect proline. The detection wavelength used in channel 2 was 570 nm, which was used to detect other amino acids. The infusion pump pressure was 0–4.2 MPa. The elution pump flow rate was 0.45 mL/min. The ninhydrin derivatization pump flow rate was 0.25 mL/min. The separation column temperature was 37 °C, and the reactor temperature was 130 °C. Flow phases: A was citric acid–lithium citrate buffer at pH 2.9; B was citric acid–lithium citrate buffer at pH 4.2; C was citric acid–lithium citrate buffer at pH 8.0; and D was 0.5 mol/L lithium hydroxide regeneration solution. FAA were confirmed by comparing the retention times and peak areas of the particular amino acid standards (WAKO-H type) with those of the components present in the samples. The following amino acids were monitored: glutamic acid (Glu), alanine (Ala), lysine (Lys), leucine (Leu), valine (Val), glycine (Gly), arginine (Arg), phenylalanine (Phe), isoleucine (Ile), proline (Pro), threonine (Thr), serine (Ser), tyrosine (Tyr), histidine (His), methionine (Met), aspartic acid (Asp), and cysteine (Cys). Three parallel tests were performed on the sample.

### 2.6. E-nose Determination

An E-nose (PEN3 Airsence, Schwerin, Germany) was applied to distinguish aroma compounds using a selective sensor array and an appropriate identification method [19]. The E-nose system consisted of 3 units: a sampling unit, a gas detection system, and pattern recognition software. In addition, the gas detector of the E-nose system consisted of 10 metal oxide semiconductor-type sensors (MOS), which had different sensitivities to each characteristic volatile compound. The main sensitive substances corresponding to each type of sensing elements are shown in Table 1.

Briefly, the minced meat sample (6 ± 0.001 g) was placed in a 50 mL headspace vessel, capped, and then allowed to stand for 30 min at an ambient temperature of 25 ± 2 °C to allow the flavor of the sample to fill the entire headspace vial. The. The flow rate in the sensor chamber was 150 mL/min, and the flow rate when the sample was measured was 100 mL/min. After cleaning the probe with filtered air for 100 s, the baseline was set (by auto-zeroing) for 10 s before the next measurement. The detection time and the cleaning time were both 240 s. In this study, the stability values of each measurement sensor were extracted and used for further data analysis. Three parallel tests were performed on the sample.

### 2.7. Statistical Analysis

All values were expressed as the mean ± standard deviation. All data were analyzed by one-way analysis of variance (ANOVA) using SPSS version 23.0 (SPSS Inc., Chicago, IL, USA). Significant differences (*p* < 0.05) among means were identified by HSD Tukey’s test. Linear Discriminant Analysis (LDA) analysis was performed using the data analysis software with the E-nose device. The data of the E-tongue were-processed using principal component analysis (PCA) with Origin 2021. Origin 2021 also was used for correlation analysis and plotting. XLSTAT 2021 was used for Multiple Factor Analysis (MFA).

## 3. Results and Discussion

### 3.1. E-Nose Analysis

LDA is a dimensionality reduction technique for supervised learning, similar to PCA, but LDA is more outstanding in supervised discrimination [20]. As shown in Figure 1A, the contribution rate of LDA main axis 1 was 75.33%, and the contribution rate of the main axis 2 was 17.4%. The cumulative variance contribution rate of the main axis 1 and the main axis 2 reached 92.7% of the original information, indicating that the two principal components of LDA analysis reflected most of the information from the original data. These results indicated that the flavor profile of fermented pork loin hams at different times differed, and the E-nose could be used to distinguish them well. Among them, the flavor profiles on the 7th, 14th, and 21st days had overlapping parts. These findings combined with the results in Table 1 showed that no significant difference was found between 7 days and 14 days, 14 days and 21 days respectively.

Figure 1B shows aroma profiles at different times. A total of ten aroma probes were used in the test. The substances sensitive to the sensors are shown in Table 1. As shown in Figure 1B, different fermentation times had apparent differences in flavor. On the seventh day of fermentation, all the aroma values were higher except for the W3S sensor. From 7 to 42 days, the aroma value of the fermented tenderloin ham gradually decreased, which meant that the content of aromatic ingredients, nitrogen oxides, hydrogen, alkanes, methane, sulfides, ethanol, and other components in the fermented tenderloin ham was gradually reduced. This indicated that the fermentation time had a significant effect on the E-nose aroma value of the fermented pork loin ham. Notably, the response values of the W1S, W1W, W2S, and W2W sensors of the fermented pork loin ham were higher than those of the other sensors, suggesting that the methyl-bearing, sulfides, and ethanol volatile compounds in fermented pork loin ham were the main flavor contributors. This might be due to the special flavor of fermented tenderloin ham. The research of Jie Shi showed that W1S, W1W, W2S, and W2W sensors were more sensitive to meat flavor substances [19]. Cumulatively, the study showed that E-nose effectively distinguished volatile compounds at different fermentation times of the fermented pork loin ham, indicating that the volatile compounds in the fermented pork loin ham changed with the change in fermentation times [21,22].

### 3.2. E-Tongue’s Analysis

The E-tongue can successfully characterize the nonvolatile flavor components in food and has higher taste sensitivity than that in the trained group [23,24]. They reflect different tastes between different samples, ranging from sourness, bitterness, saltiness, umami, astringent, after-bitterness, and after-astringency. In this test, for the reference solution, the tasteless point of sourness was −13, the tasteless point of saltiness was −6, and the tasteless point of other tastes was 0. The test results are shown in Figure 2 and Table 2.

Figure 2A shows that the umami taste of the fermented pork loin ham was more pronounced. With the increase in time, umami was consistently the highest relative to other taste receptors in the E-tongue, which might be related to the higher content of umami amino acids. The saltiness was caused by salt. However, umami also contributed to it. Harada-Padermo showed that the umami components could be used as substitutes for reducing the sodium level and retaining saltiness [25]. Bitterness was a well-known common sensory defect in dry cured ham. The bitterness of fermented pork loin ham was only second to umami in the E-tongue score. However, the bitterness in the other groups was significantly lower relative to the 0-day group. According to Xiao et al., this could be due to the use of a starter which reduced the defect flavor of ham [26]. However, the bitterness of the fermented pork loin ham was significantly increased on day 42, which might be due to excessive degradation of myosin and troponin by endogenous enzymes (cathepsin B, L and dipeptidase) [26]. The difference in sourness might be caused by *L. plantarum* SJ4, which had an acid-producing ability. The sourness values were all lower than the tasteless point −13, indicating no uncomfortableness in taste perception during the actual taste analysis. Richness could be used to characterize the residual umami taste, also known as aftertaste-umami. In the present study, the richness reached a maximum in 35 days. Some reports suggest that the increase in richness may be due to the formation of various taste substances, such as FAA, from proteolysis [27]. Overall, fermented pork loin ham consistently had the highest E-tongue umami values from 0 to 42 days relative to other taste values. It was, therefore, suggested that the umami of the fermented pork loin ham might be more pronounced across all taste values.

To better visualize the results, a principal component analysis (PCA) was carried out to visualize the results better. The results are presented in Figure 2B. Of the total variance, 94.8% was explained by the first two principal components, with PC1 and PC2 accounting for 87.9% and 6.9% of the variance, respectively. The fermented pork loin ham on day 7 was associated with bitterness. Pork loin hams at 14, 21, and 42 days were associated with umami, saltiness, astringency, and after-bitterness. The 28 and 35 day fermented pork loin hams were associated with after-astringency richness, and sourness. This indicated that the E-tongue could better distinguish samples from different fermentation periods.

### 3.3. FAA Analysis

FAA are essential substances that affect the taste of products. Larrea et al. showed that intense proteolysis occurred during the dry-cured ham processing, resulting in continuous accumulation of small peptides and FAA [28]. FAA are mainly produced by endogenous enzymes (such as cathepsins/calpains) present in raw meat and by microorganisms to promote the degradation of proteins into smaller molecules (peptides, fatty acids, and aldehydes). Joanna Stadnik indicated that after the longest examined aging time, the FAA content in the loins inoculated with a probiotic strain represented a significant increase of about 25% compared with the control sample [17]. In our previous study, *L. plantarum* SJ4 also was found to have good proteolytic properties. Therefore, it was necessary to test the FAA of fermented pork loin ham at different times to explore the correlation between FAA and volatile compounds. The results are expressed as a mean (*n* = 3).

The FAA content at different fermentation times is shown in Figure 3A. In 0–42 days, except for ILE and MET, the content of the remaining 15 FAA reached the maximum in 35 days. The amino acid levels increased rapidly between days 0-21. The studies of Yu, D suggested that adding a complex starter helped produce more FAA more quickly. This may be due to the bacterial proteolytic activity of *Lactobacillus* and *Staphylococcus* [29]. On day 35, the content of TFAA reached 12.422 mg/g. Notably, this was higher than the data of Abellán and Campus et al., which might be due to the effect of the starter [30,31]. This showed that by using a starter, more nutrients could be provided to the fermented pork loin ham. On the day 42, the contents of all amino acids showed a decreasing trend, which was mainly due to microbial metabolism being more intense than production [32]. Some studies showed that the degradation of amino acids increased flavor components [33,34]. This process might lead to the production of more volatile compound species. Except for day 0, the content of GLU was the highest among all amino acids within 7–42 days, followed by ALA. On the day 35, the content of umami amino acids, including ASP, GLU, GLY, and ALA, reached 4.694 mg/g. Quantitatively the most important FAA were Glu, Ala, and Leu. This FAA profile was mainly consistent with that reported by Lorenzo et al. [35]. In previous E-tongue tests, umami was the more prominent flavor in fermented pork loin ham. However, FAA testing suggested that this umami might be provided by umami amino acids. The content of bitter amino acids was also second only to umami amino acids. Merlo et al. speculated that FAA or interactions between compounds played a crucial role in regulating the flavor of the final product [36]. However, Zhou et al. showed that the degradation of FAA, especially VAL, LEU, and ILE, enhanced the content of 3-methyl butyric acid and 2-methyl butyric acid, resulting in the defects in ham flavor [37].

Taste activity value (TAV) is the active taste value, and its value is limited by 1 [38]. When it is lower than 1, the substance contributes little to the taste, and when it is higher than 1, it contributes more to the taste of the sample. The value of the TAV has a positive correlation with the contribution rate of taste. Figure 3B shows that the TAV value in each experimental group was small, but with the increase in time, the TAV showed a trend of first increase and then decrease; its value reached the highest value at day 35. This suggests that the flavor of the fermented pork loin ham may lead to more benefits and better absorption at day 35.

### 3.4. Analysis of Volatile Compounds Obtained by GC-MS

Appendix A shows the details of volatile compounds identified by GC-MS. Of these, 49 compounds were characterized, of which 7 were defined as food additive classes in Pubchem (https://pubchem.ncbi.nlm.nih.gov/, accessed 1 March 2022), and 35 were recorded with flavor descriptions. The types of volatile substances increased with the increase in time. A total of 11 types were found in 0 days, 24 in 7 days, 26 in 14 days, 27 in 21 days, 33 in 28 days, 27 in 35 days, and 31 in 42 days. As the end of the fermentation time on the 42nd day was reached, each volatile component maintained the state of esters (16) > aldehydes (6) > acids (4) > alcohols (3) > ketones (2).

Aldehydes are the significant by-products of lipid oxidation. However, Maillard-induced amino acid degradation also produces some aldehydes. They often have a significant impact on the characteristic flavor of dry-cured ham due to their low odor thresholds [12,39]. A total of 8 aldehyde compounds are shown in Appendix A. Decanal, nonanal and dodecanal are acyclic aliphatic aldehydes with sweet floral and fruity aromas [24]. Nonanal is generally considered the main oxidation product in dry-cured meat products and may exhibit fruit, green apple, and vegetable odors [25,40]. Phenylacetaldehyde is a Strecker aldehyde that imparts a honey-like odor to ham [36], while benzaldehyde exhibits bitter almonds, cherry, and nutty aromas.

As shown in Appendix A, ketones mainly include four compounds. Ketones are thought to result from lipid oxidation [41]. In the present study, methyl ketones, including 2-nonyl ketone and methyl nonyl ketone, were formed through chemical reactions in the presence of a large number of microorganisms, which were described as essential contributors to the complex flavor and aroma attributes of pork [12,42]. Still, except for 2-Undecanone and 2-Nonanone, the contents of other ketones were not high. Reports show that high levels of ketones are a symptom of poor quality dry-cured ham.

Alcohols are volatile compounds with sweet, fruity, or onion-like, and mushroom-like odors [39]. In this study, phenethyl alcohol was present at all fermentation stages as aromatic alcohol, which had a pleasant floral aroma. 4-Terpene alcohols are warm peppery, lighter earthy, and aged woods. With respect to the acids, the contents of octanoic acid and n-Decanoic acids remained the highest over time and might provide some cheese smell. Compared with long-chain fatty acids (Octanoic acid, n-Decanoic acid and Hexanoic acid), short-chain fatty acids such as isovaleric acid had lower thresholds due to their sensory attributes [43,44].

Esters are produced by the reaction of alcohols and carboxylic acids (or acyl-CoA) or by microbial esterase activity [45]. Long-chain acids can form long-chain esters, and long-chain esters have a slightly fatty odor. In contrast, short-chain esters have a fruity flavor [23,46], especially methyl-branched short-chain esters, which have a positive correlation with aged meat flavors [47,48]. In this study, the types and contents of esters gradually increased with the increase in time, and the type of esters were the maximum on the day 28. Their composition seemed to explain the aroma components of fermented pork loin ham. Ethyl octanoate was the most abundant compound during all periods, and the content of ethyl octanoate gradually decreased from day 7. Esters of production depending on the presence of ethanol and different acids as well as on the esterase/lipase activities of the strains [49]. Ethyl caprylate has been reported to bind to myofibrillar proteins, thereby altering the flavor of meat [50]. In general, in terms of volatile compounds, esters are probably the most important flavor compounds in fermented pork loin ham because of their higher content and variety.

The correlation of volatile compounds with FAA is shown in Appendix A. As can be seen in the figure, n-decanoic acid, benzeneacetaldehyde, tetradecanal, 2-undecanone, 2-nonanone, butanoic acid-ethyl ester, decanoic acid-ethyl ester, hexanoic acid-ethyl ester, octanoic acid-ethyl ester, dodecanoic acid-ethyl ester, tetradecanoic acid-ethyl ester, hexadecanoic acid-ethyl ester, ethyl nicotinate, ethyl 9-hexadecenoate, propanoic acid-2-hydroxy-ethyl ester, linoleic acid ethyl ester, benzeneacetic acid-ethyl ester showed a significant positive correlation with most FAA (*p* ≤ 0.01). This suggested that the increase in the content of esters represented by octanoic acid-ethyl ester in flavor substances might be related to the increase in the content of FAA with the increase in fermentation time increases. These results are of great significance to the mechanism of flavor formation.

### 3.5. MFA Analysis

MFA can be used to study relationships by simultaneously analyzing the observations, variables, and tables [51]. In this study, MFA was performed to analyze the relationship among five factors: E-nose, E-tongue, FAA, GC-MS, and fermentation time. After the analysis data were processed by dimensionality reduction, the first two factors had 65.88% (F1 and F2: 65.88%) of the variability. As shown in Figure 4, the points of each sample were projected onto the plane, and their observations were connected. The farther the observation point from the center, the better the observation point properties of the sample could distinguish it. Figure 4 shows that the E-nose aroma value of fermented pork loin ham with different fermentation time might be the key factor in distinguishing fermented pork loin ham with different fermentation times.

The RV coefficients for the correlation of variables obtained with the E-tongue, GC-MS, E-nose, and FAA in eight selected samples are shown in Table 3. It is beneficial for the dimension with the most significant explained variance. The RV coefficients had values ranging from 0 to 1, making them easy to analyze. RV coefficients >0.7 indicated a good level of differentiation [52]. The two closest tables were the E -tongue and the FAA. This indicated that the FAA in the fermented pork loin ham highly correlated with the sensory scores brought by the taste.

## 4. Conclusions

In this study, E-tongue, E-nose, GC-MS, and FAA detection were performed on pork loin ham fermented with ZF4 and SJ4 over 0–42 days. The results showed that the total FAA content reached the highest value on day 35; and the main FAA in fermented loin ham were umami amino acids, including Glu, Asp, Gly and Ala across all times. GC-MS results indicated that the flavor profile of fermented pork loin ham shifted to esters with increasing fermentation time, and the type of esters were the maximum on the 28th day. The octanoic acid-ethyl ester was the most abundant compound during all periods. In addition, the increase in the contents of esters represented by octanoic acid-ethyl ester was related to the increase of FAA contents. The FAA content also highly correlated with sensory scores of E-tongue. Therefore, we concluded that 28–35 days of fermentation time was better for the good flavor of fermented pork loin ham. The findings of this study are expected to provide a reference for the continued development of optimized fermented pork loin ham.

## Figures and Tables

**Figure 1 foods-11-01501-f001:**
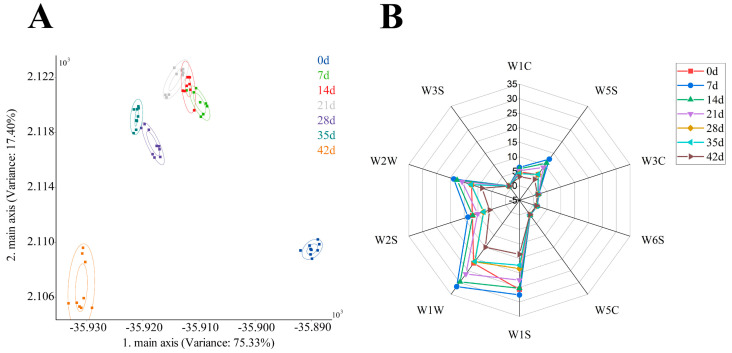
LDA analysis chart (**A**) and radar chart (**B**) of aroma at different times using the E-nose test.

**Figure 2 foods-11-01501-f002:**
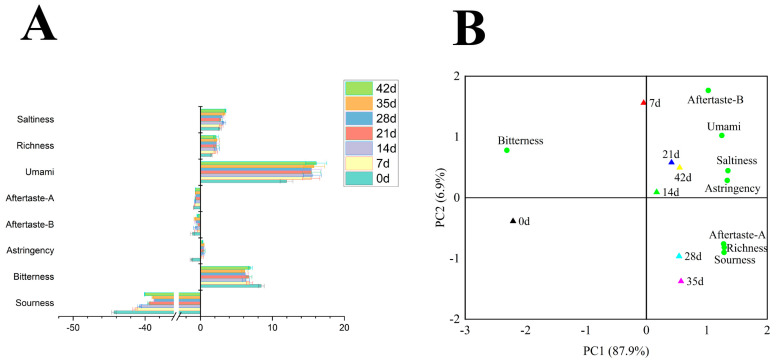
Taste value (**A**) and PCA analysis (**B**) of E- tongue at different fermentation times. PCA is expressed as the mean of flavor values.

**Figure 3 foods-11-01501-f003:**
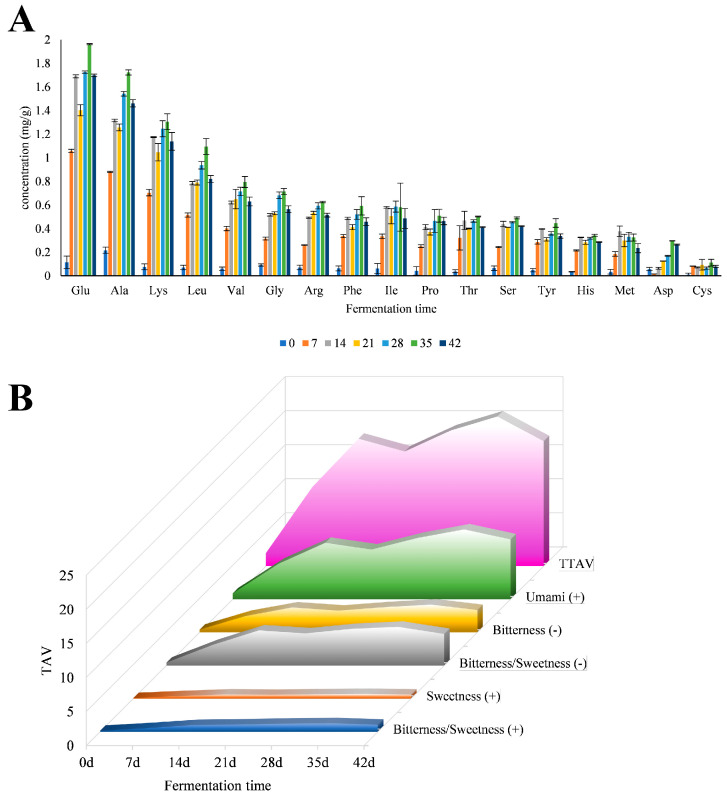
Concentrations of 17 FAA (**A**) and taste activity value (TAV) of various taste amino acids and total TAV (TTAV) at different times (**B**).

**Figure 4 foods-11-01501-f004:**
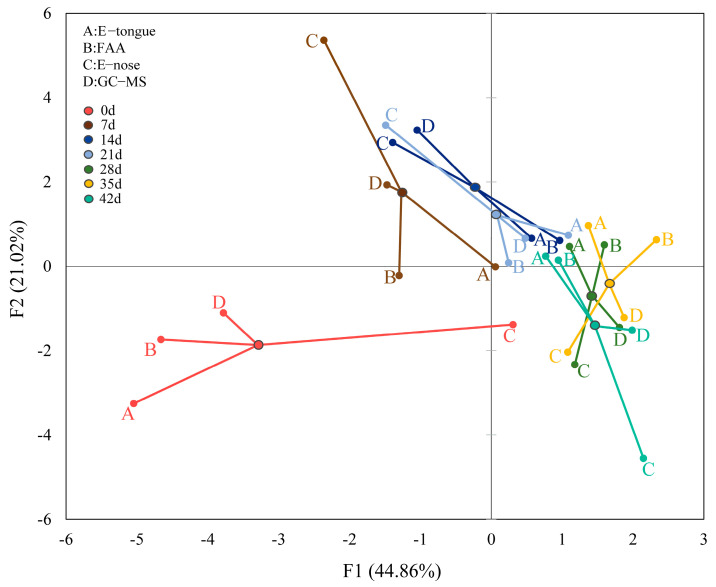
Schematic diagram of projected points of observed values.

**Table 1 foods-11-01501-t001:** Aroma values of E-nose at different fermentation times.

Sensors	Description	0 d	7 d	14 d	21 d	28 d	35 d	42 d
W1C	Sensitive to aromatic ingredients	4.79 ^bc^	6.33 ^a^	5.83 ^ab^	5.16 ^b^	4.19 ^c^	4.14 ^c^	3.14 ^d^
W5S	Sensitive to nitrogen oxides	5.94 ^c^	12.52 ^a^	10.84 ^ab^	9.00 ^b^	5.94 ^c^	6.04 ^c^	3.9 ^c^
W3C	Sensitive to aromatic ingredients	1.77 ^bc^	1.99 ^a^	1.92 ^ab^	1.83 ^b^	1.68 ^c^	1.68 ^c^	1.5 ^d^
W6S	Selective to hydrogen	1.32 ^b^	1.47 ^a^	1.36 ^b^	1.27 ^b^	1.17 ^c^	1.17 ^c^	1.06 ^d^
W5C	Sensitive to alkanes and aromatics	1.22 ^bc^	1.3 ^a^	1.28 ^ab^	1.25 ^b^	1.19 ^c^	1.19 ^c^	1.13 ^d^
W1S	Sensitive to methane	25.73 ^a^	27.56 ^a^	25.3 ^ab^	22.47 ^ab^	18.51 ^ab^	17.34 ^b^	13.57 ^b^
W1W	Sensitive to sulfides	21.79 ^bc^	31.73 ^a^	29.67 ^ab^	26.35 ^b^	21.16 ^c^	21.02 ^c^	14.95 ^d^
W2S	Sensitive to ethanol	12.02 ^ab^	13.62 ^a^	11.88 ^ab^	10.14 ^b^	8.04 ^bc^	7.92 ^bc^	5.79 ^c^
W2W	Sensitive to organic sulfides	12.32 ^c^	18.76 ^a^	17.53 ^ab^	15.71 ^b^	12.47 ^c^	12.46 ^c^	8.53 ^d^
W3S	Sensitive to alkanes	1.15 ^a^	1.14 ^ab^	1.12 ^b^	1.09 ^bc^	1.08 ^c^	1.07 ^c^	1.05 ^c^

Results are expressed as an average (*n* = 3). ^a–d^ Different superscripts in the same row indicate significant differences (*p* < 0.05).

**Table 2 foods-11-01501-t002:** Taste value of E- tongue under different fermentation times.

	0 d	7 d	14 d	21 d	28 d	35 d	42 d
Sourness	−44.33 ^g^	−41.392 ^f^	−40.77 ^e^	−39.486 ^b^	−38.732 ^a^	−38.95 ^c^	−40.06 ^d^
Bitterness	8.45 ^a^	6.816 ^b^	6.322 ^c^	6.742 ^b^	6.26 ^c^	6.156 ^c^	6.916 ^b^
Astringency	−1.25 ^c^	0.2 ^b^	0.4 ^a^	0.476 ^a^	0.46 ^a^	0.45 ^a^	0.312 ^b^
Aftertaste−B	−1.11 ^f^	−0.436 ^c^	−0.72 ^e^	−0.374 ^a^	−0.654 ^d^	−0.806 ^e^	−0.408 ^b^
Aftertaste−A	−0.93 ^e^	−0.778 ^d^	−0.748 ^d^	−0.692 ^b^	−0.656 ^a^	−0.692 ^c^	−0.71 ^c^
Umami	11.96 ^b^	15.398 ^a^	15.558 ^a^	15.43 ^a^	15.446 ^a^	15.772 ^a^	16.108 ^a^
Richness	1.54 ^b^	2.14 ^a^	2.286 ^a^	2.22 ^a^	2.236 ^a^	2.318 ^a^	2.176 ^a^
Saltiness	2.5 ^f^	2.89 ^e^	3.264 ^c^	2.848 ^e^	3.026 ^d^	3.36 ^b^	3.496 ^a^

Results are expressed as an average (*n* = 3). ^a–f^ Different superscripts in the same row indicate significant differences (*p* < 0.05).

**Table 3 foods-11-01501-t003:** RV coefficients of E-nose, E-tongue, FAA, and GC-MS.

	E-Nose	E-Tongue	FAA	GC-MS
E-nose	1.000	0.116	0.209	0.491
E-tongue	0.116	1.000	0.875	0.574
FAA	0.209	0.875	1.000	0.586
GC-MS	0.491	0.574	0.586	1.000

## Data Availability

Not applicable.

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
