# Peer review of "Studies on Flavor Compounds and Free Amino Acid Dynamic Characteristics of Fermented Pork Loin Ham with a Complex Starter"

_foods, 2022, doi:10.3390/foods11101501_

Round 1

Reviewer 1 Report

Title

Authors should correct the article title with the help of a native speaker. It is unreadable and unclear, too much information the authors wanted to convey with too few words. flavor files? - what is that?

Abstract

line 17 – give names of this free amino acids if you gave names of umami amino acids

line 18-19 – “Electronic tongue (E-tongue) analysis indicated that the umami taste of fermented pork loin ham was more evident in all taste values.” – please explain and rewrite this sentence

line 25 - to distinguish fermented pork loin ham but among what? all period samples? be specific

Introduction

line 46 - The authors write that several studies have shown no significant difference in sensory evaluation between 60 and 90 days in 30, 60, and 90 days of naturally fermented tenderloin ham. But they cited only one literature item. Please rewrite sentence or add more literature. And what did the authors mean by this?

line 46-48 – this sentence needs improvement, is incomprehensible

Authors should extend the introduction by recent advances in studies about flavor of fermented meats.

The authors do not provide a rationale for undertaking the study - what prompted them to do the study in this setting? Why these microorganisms were chosen?

Materials and Methods

72-74 - The authors provide no information about the origin of the meat - what breeds of animals were used, how old they were slaughtered, what they ate, and for how long? How many pieces of meat were used in the study? What spices were used? All of this information is important to the results of this research. Please provide raw material specifications.

line 84-85 – please check this sentence, “clear liquid was filtered in a 40 ℃ water bath for 10 minutes” ? what do you mean?

line 84, 94 - How was the sample taken and from which part of the meat? Was it always taken from the same part at each fermentation period? Was the sample shredded or added in one piece?

line 94-95- On what basis were the extraction parameters chosen and what were the volatile compounds extracted with?

line 123-124- unique headspace vessel was opened or closed during 30 minutes? in what temperature?

line 83-84, 123 - what sensors have been chosen for each-tongue and e-nose determination?  this is crucial for research

line 130 and 131- name this software

Results – and Discussion?

authors do not provide figures and tables which makes it impossible to review this section

Discussion – or Conclusions?

The entire section must be rewritten. The authors repeat information regarding the results obtained. They state obvious observations. There are no real conclusions of the study.

line 290-291- please delete this sentence, its repetition of given informations

line 293-294 - “The E-tongue test results also showed that the umami taste of fermented pork loin ham was more pronounced.” - more pronounced than what?

line 294-295 - it's obvious, you don't have to do a research to find out this

line 300-301 – “MFA analysis indicated that the E-nose sensor value of fermented pork loin ham might be the key factor to distinguish fermented pork loin ham.” - Any statistical analysis will show that the sample from day 0 is different from the sample from day 42 of fermentation. It is enough to observe the change in color and texture with the human senses to distinguish fresh meat from fermented meat.

Author Response

Dear reviewers:

We appreciate the opportunity to revise our manuscript. Moreover, we would like to thank the reviewers for careful and thorough reading of this manuscript and for the thoughtful comments and constructive suggestions, which help to improve the quality of our manuscript. A point-by-point response to the Reviewers’ comments is below.

Note: Due to too many changes, it is not possible to mark the changes in the revised manuscript.

But the main changes are as follows:

  1. Reorganize the introduction part:
  2. The results and discussion section focus on the presentation of conclusions:

1) Both the E-nose and E-tongue showed good discrimination ability for fermented tenderloin ham with different fermentation time, which was crucial in cases with large clusters.

2) E-nose aroma value of fermented pork loin ham with different fermentation time might be the key factor in distinguishing fermented pork loin ham with different fermentation time

3) The increase of esters represented by octanoic acid-ethyl ester may be related to the increase of free amino acids with the increase of fermentation time.

4) The main free amino acids in fermented tenderloin ham are umami amino acids, they may be the reason for the prominent umami taste of electronic tongue. Because the RV coefficient (0.875) indicates that free amino acids are highly correlated with the sensory score of the E-tongue

  1. Optimize the graphics of the article (font, clarity, etc.).
  2. According to the comments of the reviewers, the details of the article are revised.
  3. Language editing has been done.

Point 1: Authors should correct the article title with the help of a native speaker. It is unreadable and unclear, too much information the authors wanted to convey with too few words. flavor files? - what is that?

Response: Thanks for your suggestion, the language editing has been done as you suggestion. We have attached a language editing certificate. In the revised version, we have revised the title: Studies on flavor compounds and free amino acid dynamic characteristics of fermented pork loin ham with a complex starter However, the initial idea was to use the evolution of VOC and FAA information in fermented pork loin ham, including volatile compounds, flavor values, etc., to gain a clearer understanding of the changing trends and correlations of indicators during the fermentation process.

Point 2: line 17 – give names of this free amino acids if you gave names of umami amino acids

Response: Thanks for your suggestion. Details have been added in our revised manuscript

Point 3: line 18-19 – “Electronic tongue (E-tongue) analysis indicated that the umami taste of fermented pork loin ham was more evident in all taste values.” – please explain and rewrite this sentence

Response: Thanks for your suggestion. Details have been rewrite in our revised manuscript: line 199-200“With the increase in time, umami is consistently the highest relative to other taste receptors in the E-tongue, which might be related to the higher content of umami amino acids.”.

Point 4: line 25 - to distinguish fermented pork loin ham but among what? all period samples? be specific

Response: Thanks for your suggestion. Details have been added in our revised manuscript: Line 24-25 “both the electronic nose and electronic tongue showed good discrimination ability for fermented tenderloin ham with different fermentation time, which was crucial in cases with large clusters”.

Point 5: line 46 - The authors write that several studies have shown no significant difference in sensory evaluation between 60 and 90 days in 30, 60, and 90 days of naturally fermented tenderloin ham. But they cited only one literature item. Please rewrite sentence or add more literature. And what did the authors mean by this?

Response: Thanks for your suggestion. Details have been changed in our revised manuscript: line43-45 “Seong et al. reported that the flavor of tenderloin ham varies significantly in the short term, but then tends to be consistent [8]. Therefore, we hope to explore the dynamics of tenderloin ham through some modern sensory technologies.”

  1. Seong, P.N., K.M. Park, G.H. Kang, S.H. Cho, B.Y. Park, and B. Hoa Van, The Impact of Ripening Time on Technological Quality Traits, Chemical Change and Sensory Characteristics of Dry-cured Loin. Asian-Australasian Journal of Animal Sciences, 2015. 28(5): p. 677-685.doi:10.5713/ajas.14.0789.

Point 7: Authors should extend the introduction by recent advances in studies about flavor of fermented meats. The authors do not provide a rationale for undertaking the study - what prompted them to do the study in this setting? Why these microorganisms were chosen?

Response: Thanks for your reading and attention! In the revised version, we have reedited this part: line34-42 “As we all know, as a geographical indi-cation product, the flavor quality of ham is inseparable from environmental microorganisms. Some reports have summarized the relationship between microorganisms and ham flavor formation [3, 4]. Generally speaking, it is because they can degrade proteins and lipids to form micromolecules substance. Therefore, in some recent studies, researchers have gained interest in fermented meat using beneficial microbes from local specialty meat products as a starter, giving fermented meat more local characteristics [5, 6]. According to Laranjo, starters, especially Lactobacillus and Staphylococci, can improve the safety of fermented meat products, help standardize product characteristics, and shorten the aging time [7].”

Point 9: 72-74 - The authors provide no information about the origin of the meat - what breeds of animals were used, how old they were slaughtered, what they ate, and for how long? How many pieces of meat were used in the study? What spices were used? All of this information is important to the results of this research. Please provide raw material specifications.

Response: Thanks for your suggestion. Details have been added in our revised manuscript: line 72-81“The diet of landrace pigs was slightly adjusted in different stages, and the main ingredients were corn and wheat bran. Landrace pigs were slaughtered after being fed about 250kg of diet, and the tenderloin was sold by Fuzhiyuan Group Co., Ltd. Spices (food-grade) were purchased from a supermarket in Guiyang City, and the casings were purchased from the Shenguan Group. The pork loin was cut into about 500-g per pork loin cubes equally (n=21). The loin’s surface was rubbed with a formula containing 3% salt(w/w), 0.01% sodium nitrite, 0.2% sodium erythorbate, 0.5% sodium tripolyphosphate, 1% sugar, and 1% glucose, and kept at 4℃ for 24 h to allow the seasoning mixture to distribute into the meat. The prepared bacterial suspension was inoculated into the meat (107 CFU/g), with a ZF4 to SJ4 ratio of 1:1”

Point 10: line 84-85 – please check this sentence, “clear liquid was filtered in a 40 ℃ water bath for 10 minutes” ? what do you mean?

Response: Thanks for your suggestion. Details have been added in our revised manuscript: “The samples (60 g) were added to 240 mL of deionized water, followed by a 40℃-water bath for 10 min before the liquid was filtered”.

Point 11: line 84, 94 - How was the sample taken and from which part of the meat? Was it always taken from the same part at each fermentation period? Was the sample shredded or added in one piece?

Response: Thanks for your suggestion. Details have been added in our revised manuscript. Line77、87: “The pork loin was cut into about 500-g per pork loin cubes equally(n=21) …… Each time, 3 fermented tenderloin hams of the same period were minced and mixed, then put into sterile vacuum bags and stored at -20°C until testing.”

Point 12: line 94-95- On what basis were the extraction parameters chosen and what were the volatile compounds extracted with?

Response: Thanks for your reading and attention! In the revised version, we corrected these errors: line10-103: “Initially, 5 g of the sample was placed in a 20-mL headspace sample bottle and equilibration of the headspace was done maintaining the vial at 60°C during 30min.”

Point 13: line 123-124- unique headspace vessel was opened or closed during 30 minutes? in what temperature?

Response: Thanks for your suggestion. Details have been added in our revised manuscript: line 141“Briefly, the minced meat sample (6 ± 0.001 g) was placed in a 50 mL headspace vessel, capped, and then allowed to stand for 30 min at ambient temperature of 25 ± 2 °C to allow the flavor of the sample to fill the entire headspace vial.”

Point 14: line 83-84, 123 - what sensors have been chosen for each-tongue and e-nose determination?  this is crucial for research

Response: Thanks for your suggestion. Details have been added in our revised manuscript: line 191-193 “They reflect different tastes between different samples, ranging from sour, bitter, salty, umami, astringent, after-bitterness, and after-astringency.” For E-tongue sensors. Line 186 Table 1 describes the sensors of the E- nose

Point 15: line 130 and 131- name this software

Response: Thanks for your suggestion. Details have been added in our revised manuscript: line 109 The Xcalibur software……

Point 16: authors do not provide figures and tables which makes it impossible to review this section

Response: Thanks for your suggestion. In the revised version, we corrected these errors. All graphic elements are inserted into the text.

Point 17: The entire section must be rewritten. The authors repeat information regarding the results obtained. They state obvious observations. There are no real conclusions of the study.

Response: Thanks for your reading and attention! In the revised version, we have re-edited this section. Due to the large number of changes, it cannot be marked, so the main results are as follows:

1、both the E-nose and E-tongue showed good discrimination ability for fermented tenderloin ham with different fermentation time, which was crucial in cases with large clusters.

2、E-nose aroma value of fermented pork loin ham with different fermentation time might be the key factor in distinguishing fermented pork loin ham with different fermentation time

3、the increase of esters represented by octanoic acid-ethyl ester may be related to the increase of free amino acids with the increase of fermentation time.

4、The main free amino acids in fermented tenderloin ham are umami amino acids, they may be the reason for the prominent umami taste of electronic tongue. Because the RV coefficient (0.875) indicates that free amino acids are highly correlated with the sensory score of the E-tongue

Point 18: line 290-291- please delete this sentence, its repetition of given informations

Response: Thanks for your reading and attention! In the revised manuscript we had removed this sentence.

Point 19: line 293-294 - “The E-tongue test results also showed that the umami taste of fermented pork loin ham was more pronounced.” - more pronounced than what?

Response: Thanks for your suggestion. Details have been added in our revised manuscript. Line355:"The E-tongue test results also showed that the umami is consistently the highest relative to other taste sensors with the increase in time.”

Point 20: line 294-295 - it's obvious, you don't have to do a research to find out this

Response: Thanks for your suggestion. In the revised manuscript we had deleted this sentence.

Point 21: line 300-301 – “MFA analysis indicated that the E-nose sensor value of fermented pork loin ham might be the key factor to distinguish fermented pork loin ham.” - Any statistical analysis will show that the sample from day 0 is different from the sample from day 42 of fermentation. It is enough to observe the change in color and texture with the human senses to distinguish fresh meat from fermented meat.

Response: Thanks for your suggestion. In the revised version, we have reedited this part: line 344 “MFA can be used to study relationships by simultaneously analyzing the observations, variables, and tables [51]. In this study, the points of each sample were projected onto the plane, and their observations were connected. The farther the observation point from the center, the better the observation point properties of the sample could distinguish it. This result shows that the E-nose aroma value of fermented pork loin ham with different fermentation time might be the key factor in distinguishing fermented pork loin ham with different fermentation time.”

  1. dos Santos Scholz, M.B., C.S. Good Kitzberger, S.H. Prudencio, and R.S. dos Santos Ferreira da Silva, The typicity of coffees from different terroirs determined by groups of physico-chemical and sensory variables and multiple factor analysis. Food Research International, 2018. 114: p. 72-80.doi:10.1016/j.foodres.2018.07.058.

Reviewer 2 Report

The manuscript Foods-1678515 “Studies on flavor files and free amino acids of fermented tenderloin ham with the complex starter at different times”

The manuscript I had access do not have any figures or tables, that hinder a more serious revision.

The article has a complex experimental analysis of the cured fermented loin. There are several aspects that can be clarified.

A general comment. I am not familiar with the designation of “flavor files”. I understand that the electronic approach used in this study produces a huge number of files, but the aim of the work was to study the evolution of the sensory characteristics, traits, or attributes of the product. Can the authors explain what is their understanding of these “files”?

Please review the Latin names of bacteria. Upper case in the genus name, lower case in the second part of the name; please do not use italic for non-Latin names, like “lactic acid bacteria”.

Please consider allocating the identification of the volatile compounds (lines 129-130) in the 2.4 section. It is a procedure of this method, not a statistical analysis

Line 52 – Please consider changing the abbreviation of free amino acids to FAA. The FFA authors used in the article is commonly used for” free fatty acids”.

Item 2.2. In material and methods, it would be interesting to know the experiment design in more detail, namely if the processing of the loins was repeated, and if yes, how many repetitions.

After the fermentation period at 30ºC, how were the cured loins dried. Was is kept at 30ºC?. Please clarify

Item 2.7. Please consider allocating the identification of the volatile compounds (lines 129-130) in the 2.4 section. Also, the identification of compounds from electronic devices is not statistical produce by authors use to validate the research hypothesis. Please consider maintaining in this section only the ANOVA (?) used to detect differences.

What was compared to detect differences? (in the results it is possible to understand that it was the curing times that were compared, but it should be clarified in the MM section, and based on which repetitions?

Without the graphic elements, it is very difficult to understand the presentation of results.

Author Response

Dear reviewers:

We appreciate the opportunity to revise our manuscript. Moreover, we would like to thank the reviewers for careful and thorough reading of this manuscript and for the thoughtful comments and constructive suggestions, which help to improve the quality of our manuscript. A point-by-point response to the Reviewers’ comments is below.

Note: Due to too many changes, it is not possible to mark the changes in the revised manuscript.

But the main changes are as follows:

  1. Reorganize the introduction part:
  2. The results and discussion section focus on the presentation of conclusions:

1) Both the E-nose and E-tongue showed good discrimination ability for fermented tenderloin ham with different fermentation time, which was crucial in cases with large clusters.

2) E-nose aroma value of fermented pork loin ham with different fermentation time might be the key factor in distinguishing fermented pork loin ham with different fermentation time

3) The increase of esters represented by octanoic acid-ethyl ester may be related to the increase of free amino acids with the increase of fermentation time.

4) The main free amino acids in fermented tenderloin ham are umami amino acids, they may be the reason for the prominent umami taste of electronic tongue. Because the RV coefficient (0.875) indicates that free amino acids are highly correlated with the sensory score of the E-tongue

  1. Optimize the graphics of the article (font, clarity, etc.).
  2. According to the comments of the reviewers, the details of the article are revised.
  3. Language editing has been done.

Point 1: The manuscript I had access do not have any figures or tables, that hinder a more serious revision.

Response: Thanks for your suggestion. Details have been added in our revised manuscript

Point 2: A general comment. I am not familiar with the designation of “flavor files”. I understand that the electronic approach used in this study produces a huge number of files, but the aim of the work was to study the evolution of the sensory characteristics, traits, or attributes of the product. Can the authors explain what is their understanding of these “files”?

Response: Thanks for the suggestion, in the revised version, we have revised the title: Studies on flavor compounds and free amino acid dynamic characteristics of fermented pork loin ham with a complex starter However, the initial idea was to use the evolution of VOC and FAA information in fermented pork loin ham, to gain a clearer understanding of the changing trends and correlations of indicators during the fermentation process.

Point 3: Please review the Latin names of bacteria. Upper case in the genus name, lower case in the second part of the name; please do not use italic for non-Latin names, like “lactic acid bacteria”.

Response: Thanks for your suggestion. In the revised version, we corrected these errors.

Point 4: Please consider allocating the identification of the volatile compounds (lines 129-130) in the 2.4 section. It is a procedure of this method, not a statistical analysis

Response: Thanks for your suggestion. In the revised version, we corrected these errors.

Point 5: Line 52 – Please consider changing the abbreviation of free amino acids to FAA. The FFA authors used in the article is commonly used for” free fatty acids”.

Response: Thanks for your suggestion. In the revised version, we corrected these errors.

Point 6: Item 2.2. In material and methods, it would be interesting to know the experiment design in more detail, namely if the processing of the loins was repeated, and if yes, how many repetitions.

Response: Thanks for the suggestion, the process was repeated 3 times. In the revised version, I had made the following changes: “The samples were taken every 7 days until the end of fermentation on day 42. Every time, the fermented pork loins were minced, vacuum packed in sterile bags and frozen at −20 °C, and using 3 loins as replicates (n=21)”.

Point 7: After the fermentation period at 30ºC, how were the cured loins dried. Was is kept at 30ºC?. Please clarify

Response: Thanks for your suggestion, Details have been added in our revised manuscript. Line82-84 “hung in a constant temperature and constant humidity fermentation cabinet. The fermentation cabinet was set to 30°C and 85% RH with stable circulating air to allow the starter to work……”

Point 8: Item 2.7. Please consider allocating the identification of the volatile compounds (lines 129-130) in the 2.4 section. Also, the identification of compounds from electronic devices is not statistical produce by authors use to validate the research hypothesis. Please consider maintaining in this section only the ANOVA (?) used to detect differences.

Response: Thanks for your suggestion. Details have been added in our revised manuscript. line149-151 “All data were analyzed by one-way analysis of variance (ANOVA) using SPSS version 23.0 (SPSS Inc., Chicago, USA). Significant differences (P < 0.05) among means were identified by HSD Tukey's test.”

Point 9: What was compared to detect differences? (in the results it is possible to understand that it was the curing times that were compared, but it should be clarified in the MM section, and based on which repetitions?

Response: Thanks for your suggestion. Details have been added in our revised manuscript: “The samples were taken every 7 days until the end of fermentation on day 42 to compare the dynamic evolution of flavor and free amino acids at different fermentation times. Every time, the fermented pork loins were minced, vacuum packed in sterile bags and frozen at −20 °C, and using 3 loins as replicates (n=21)”.

Point 10: Without the graphic elements, it is very difficult to understand the presentation of results.

Response: Thanks for your suggestion. In the revised version, we corrected these errors. All graphic elements are inserted into the text.

Round 2

Reviewer 1 Report

line 90- „flavor files” – what is this? there is no such phrase. please change this words in whole manuscript

line 137-138- “Initially, 5 g of the sample was placed in a 20 -mL headspace sample bottle and equilibration of the headspace was done maintaining the vial at 60 °C during 30min” – Why didn't you use NaCl solution? how long extraction lasts and in what parameters (if you did SPME)? On what basis were the equilibration and extraction parameters chosen? –previous studies, preliminary studies or other?

line 142- “The sample was injected without splitting.” – did you do SPME analysis or direct headspace air injection?

2.7 Statistical analysis - you didn't mention that you also performed a PCA analysis in the case of e-tongue

Figure 1 and Figure 2 and Figure 5 - you need to enlarge the font of words and number of days because now they're blurred and hard to read

Figure 4 - Absolutely nothing can be seen on a set of images, so I suggest to add each image individually in supplementary material.

Conclusion

The authors do not state the conclusion of what the addition of the microbial starters SJ4 and ZF4 gave. This is important because the authors emphasized at the beginning of the article that they used selected microorganisms.

Author Response

Comments of Review1

Dear reviewers:

We appreciate the opportunity to revise our manuscript again. Moreover, we would like to thank the reviewers for careful and thorough reading of this manuscript and for the thoughtful comments and constructive suggestions, which help to improve the quality of our manuscript. A point-by-point response to the Reviewers’ comments is below.

Point 1: line 90- „flavor files” – what is this? there is no such phrase. please change this words in whole manuscript

Response: Thanks for your suggestion. Details have been changed in our revised manuscript.

Point 2: line 137-138- “Initially, 5 g of the sample was placed in a 20 -mL headspace sample bottle and equilibration of the headspace was done maintaining the vial at 60 °C during 30min” – Why didn't you use NaCl solution? how long extraction lasts and in what parameters (if you did SPME)? On what basis were the equilibration and extraction parameters chosen? –previous studies, preliminary studies or other?

Response: Thanks for your suggestion. Details have been added in our revised manuscript.

Line 111-113: “Initially, 5 g of the sample was placed in a 20-mL headspace sample bottle and equilibration of the headspace was done maintaining the vial at room temperature (25±2°C) during 30min. As described Perea-Sanz with minor modifications [18].”

As reported by Perea-Sanz and Belloch et al. [16,18], NaCl solution is not used for extraction. In the experiments of the team, NaCl was also not used. So I didn't use NaCl solution.

Line113-117 added SPME program: “The 50/30 μm DVB/Carboxen/PDMS fiber (Supelco, Bellefonte, PA) was headed into the SPME device, which was inserted in the vial and shaken at 50°C for 30 min to ex-tract and absorb the volatile components. Then, it was desorbed in 7 min at 250°C into the GC inlet with the automatic autosampler.”

16            Belloch, C., A. Neef, C. Salafia, J.J. Lopez-Diez, and M. Flores, Microbiota and volatilome of dry-cured pork loins manufactured with paprika and reduced concentration of nitrite and nitrate. Food Research International, 2021. 149.doi:10.1016/j.foodres.2021.110691.

  1. Perea-Sanz, L., R. Montero, C. Belloch, and M. Flores, Microbial changes and aroma profile of nitrate reduced dry sausages during vacuum storage. Meat Science, 2019. 147: p. 100-107.doi:10.1016/j.meatsci.2018.08.026.

Point 3: line 142- “The sample was injected without splitting.” – did you do SPME analysis or direct headspace air injection?

Response: Thanks for your suggestion. We were used SPME analysis. Details have been changed in our revised manuscript. Line 120: “The analysis was performed in the spliless mode.”

Point 4: 2.7 Statistical analysis - you didn't mention that you also performed a PCA analysis in the case of e-tongue

Response: Thanks for your suggestion. Details have been added in our revised manuscript.

Point 5: Figure 1 and Figure 2 and Figure 5 - you need to enlarge the font of words and number of days because now they're blurred and hard to read

Response: Thanks for your suggestion. the font of words and number of days have been enlarged in our revised manuscript.

Point 6: Figure 4 - Absolutely nothing can be seen on a set of images, so I suggest to add each image individually in supplementary material.

Response: Thanks for your suggestion. Details have been changed in our revised manuscript.

Point 7: The authors do not state the conclusion of what the addition of the microbial starters SJ4 and ZF4 gave. This is important because the authors emphasized at the beginning of the article that they used selected microorganisms.

Response: Thanks for your suggestion. Details have been changed in the conclusion section. Line 380-392: “In this study, E-tongue, E-nose, GC-MS, and FAA detection were performed on fermented pork loin ham with ZF4 and SJ4 during 0–42 days. The test results showed that the total FAA content reached the highest value on the 35th day, and the content of umami amino acids was the highest, including ASP, GLU, GLY, and ALA. For volatile com-pounds identified, the flavor profile of fermented pork loin ham shifted to esters with increasing fermentation time. Octanoic acid-ethyl ester was the most abundant compound during all periods. Most importantly, with the increase in fermentation time, the increase in the contents of esters represented by octanoic acid-ethyl ester might be related to the in-crease of in the contents FAA. FAA also highly correlated with sensory scores of E-tongue. These results are of great significance to the mechanism of flavor formation. In general, this study find that fermented pork loin ham had good flavor and good overall quality when it was fermented for 28-35 days. The findings of this study are expected to provide a reference for the continued development of optimized fermented pork loin ham.”

In this study, we focused on the characteristics of different times, not the characteristics of the strains, so we have no way to concluded about the characteristics of the strains. The characteristics of the strain are as follows:

  1. plantarum SJ4 was preserved by the China Center of Industrial Culture Collection (strain number: 20-523-1634-2388). It has proteolytic properties.
  2. simulans ZF4 (The accession number of the gene sequence on Genbank is ON193993) was screened with nitrate reductase as the inspection index and was preserved in the laboratory. ZF4 had been tested for safety, including biofilm testing, pathogenic gene testing, antibiotic testing, etc. In addition, some reports that Staphylococcus is considered to participate in the development of esters. These reports about strains in another paper that we will publish soon.
